# Morphological Effects of Strain Localization in the Elastic Region of Magnetorheological Elastomers

**DOI:** 10.3390/ma15238565

**Published:** 2022-12-01

**Authors:** Mohd Aidy Faizal Johari, Saiful Amri Mazlan, Nur Azmah Nordin, Seung-Bok Choi, Siti Aishah Abdul Aziz, Shaari Daud, Irfan Bahiuddin

**Affiliations:** 1Engineering Materials & Structures (eMast) Ikhoza, Malaysia-Japan International Institute of Technology (MJIIT), Universiti Teknologi Malaysia, Kuala Lumpur 54100, Malaysia; 2Department of Mechanical Engineering, The State University of New York, Korea (SUNY Korea), 119 Songdo Moonhwa-Ro, Yeonsu-Gu Incheon, Incheon 21985, Republic of Korea; 3Department of Mechanical Engineering, Industrial University of Ho Chi Minh City (IUH), 12 Nguyen Van Bao Street, Go Vap District, Ho Chi Minh City 70000, Vietnam; 4Faculty of Applied Sciences, Universiti Teknologi MARA (UiTM) Cawangan Pahang, Bandar Tun Abdul Razak Jengka 26400, Malaysia; 5Department of Mechanical Engineering, Vocational College, Universitas Gadjah Mada, Yogyakarta 55281, Indonesia

**Keywords:** durability, elasticity, microplasticity, morphology, shear band, strain localization, viscoelastic materials

## Abstract

Strain localization is a significant issue that poses interesting research challenges in viscoelastic materials because it is difficult to accurately predict the damage evolution behavior. Over time, the damage mechanism in the amorphous structure of viscoelastic materials leads to subsequent localization into a shear band, gradually jeopardizing the materials’ elastic sustainability. The primary goal of this study is to further understand the morphological effects and the role of shear bands in viscoelastic materials precipitated by strain localization. The current study aims to consolidate the various failure mechanisms of a sample and its geometry (surface-to-volume ratio) used in torsional testing, as well as to understand their effects on stress relaxation durability performance. A torsional shear load stress relaxation durability test was performed within the elastic region on an isotropic viscoelastic sample made of silicon rubber and a 70% weight fraction of micron-sized carbonyl iron particles. Degradation was caused by a shear band of localized plasticity that developed microscopically due to stress relaxation durability. The failure pattern deteriorated as the surface-to-volume ratio decreased. A field-emission scanning electron microscope (FESEM) and a tapping-mode atomic force microscope (AFM) were used for further observation and investigation of the sample. After at least 7500 cycles of continuous shearing, the elastic sustainability of the viscoelastic materials microstructurally degraded, as indicated by a decline in stress performance over time. Factors influencing the formation of shear bands were observed in postmortem, which was affected by simple micromanipulation of the sample geometry, making it applicable for practical implementation to accommodate any desired performance and micromechanical design applications.

## 1. Introduction

In practice, an elastic product or part can withstand the distorting influence and deformation of a force and immediately return to its original state after the force is removed. When viscoelastic materials are introduced, they retain their original ability but exhibit viscous and elastic characteristics when deformed. The elastic ability, however, is dependent on and limited to a single designated maximum force. This limitation causes the material to deteriorate over time as the force becomes dynamic and localized. The material itself loses its ability to sustain elasticity due to microstructural degradation, requiring regular part replacement and maintenance. To address this issue, a viscoelastic material known as magnetorheological elastomer (MRE) with long-lasting elastic properties that change in response to external stimuli has been introduced [1,2,3]. This material has been labeled a smart material due to its combination of an elastomeric matrix and implanted ferromagnetic carbonyl iron particles (CIPs) that respond to external magnetic stimulation. The matrix has an amorphous molecular structure, and the organization of the molecular chain is typically composed of randomly ordered, entangled, sprightly molecules. The cross-linkages formed during the curing process determine the material’s elasticity and deformation limitations. These cross-linkages are an Important structural component, as they impart the high elastic properties of MRE while also stiffening the material system to prevent a viscous dominant response [4]. The interaction of the elastic matrix and the embedded particles responsive to magnetic assortment produces an MRE with a sustainable elastic. This means that MRE can sustain elasticity toward a range of localized deformations, eliminating the need for regular part replacement.

However, MRE degradation at the molecular level because of localized deformation poses a threat, especially over long periods of time. This has prompted many studies, as reported in the literature [5], to conduct durability research, which has long been the subject of unfair debate. One of the best phenomena to study for localized deformation is stress relaxation, which describes how materials relieve stress under the constant strain generated in the structure. Although stress relaxation has been widely used to characterize the time-dependent viscoelastic properties of polymer materials, MRE research is still in its early stages. To date, only a few studies [6,7,8,9] have focused on MRE stress relaxation, and the majority have reported that the molecular structure was strained for a finite time during the stress relaxation process, resulting in a strained microplastic. Following that, microplasticity deformation occurred because of a series of local reorganizations. However, it has become evident that the elastic deformation of MRE is far more complicated. MRE has a distinct molecular environment that influences its response to macroscopic loading conditions due to its sustainable elastic characteristic from tunable external stimulation. This distinction has several implications, including microplasticity at extremely small strains [10] within the elastic region. Consequently, MRE experiences an intriguing phenomenon known as strain localization, in which the materials deform plastically in extremely small regions of the elastic domain known as shear bands [11].

Several studies [12,13,14,15,16,17] have been conducted to investigate the shear band formation in solid amorphous materials similar to MRE. This includes the shear band’s orientation relative to the principal stress axis, which varies significantly depending on the external loading conditions [18]. In general, the shear band is the primary feature that regulates the plastic deformation process. However, direct research on the shear band and its properties is uncommon, as it is considered extremely narrow [19]. The factors that govern the shear banding process in certain materials remain unknown. It is unknown how much the geometry of shear band patterns is influenced by the chosen system’s boundary conditions and/or material properties [20]. Despite advances in materials science and the introduction of smart viscoelastic materials, such as MRE, the fundamental morphological effects caused by microstructural degradation are still unclear [14]. Therefore, given the scarcity of evidence-based literature on this phenomenon, this study offers an opportunity to gain a better understanding of the microscopic effects of strain localization on MRE. This is the main technical contribution of this work. An evaluation is performed on a much narrower area of an MRE sample than that previously reported [21] to make a comparison of the surface-to-volume ratio. Concomitantly, a field-emission scanning electron microscope (FESEM) and an atomic force microscope (AFM) are used in this study to investigate stress relaxation and strain localization phenomena in MRE. The morphological effect of micromanipulation on the molecular structure arrangement is highlighted as a pristine attempt is made to perform a stress relaxation test on the MRE sample closest to its state of rest, known as the equilibrium condition. The characteristics and processes derived from this achievement are critical in understanding the sustainable elastic behavior of MRE and its deformation zone.

The primary goal of this study is to further understand the morphological effects and the role of shear bands in viscoelastic materials precipitated by strain localization. This study also aims to consolidate the various failure mechanisms of the sample and its geometry (surface-to-volume ratio) used in torsional testing, as well as to understand their effects on stress relaxation durability performance. Although the general form of the test does not show a significant change, the changes in the specimen size (surface-to-volume ratio) cannot be ignored. To date, the effect of these variables has not been well-understood. For morphological aspects, data accuracy and consistency are critical. Understanding the variability in the failure mechanism is essential for evaluating durability performance. This study investigates the effects of test geometry, specifically the surface-to-volume ratio, on the reported reduction stresses, assuming that the material’s failure process of stress relaxation remains consistent. The effect of different surface-to-volume ratios is evaluated morphologically for various mechanisms. Consequently, this study encourages innovation and significantly increases the number of scientific studies and technological capabilities in accordance with the 9.5 sustainable development (SDG) agenda.

## 2. Materials and Methods

### 2.1. Production of the Test Sample

Process parameters that are varied or altered during the fabrication process can easily manipulate or affect the material properties of MRE. As a result, special consideration must be given to the fabrication process when producing MRE samples so that the full potential of MRE can be realized. Previous researchers have reported on related studies regarding the effect of particle fraction and MRE arrangement [3,22,23,24,25,26,27]. In this study, a sample was fabricated using a similar established method from previous studies [4,8,17,28], with the help of a requisite cylindrical closed mold. The fabrication process began with the mixing of silicone rubber and particles. This was accomplished by mixing the soft CIPs (d50 = 3.8–5.3 µm, CC grade, supplied by BASF, Ludwigshafen, Germany) via mechanical stirring into room-temperature-vulcanized (RTV) silicone rubber (NS625tds), supplied by Nippon Steel Co., Ltd., Tokyo, Japan, at a controlled speed of 200 rpm and a room temperature of 25 °C. The mixtures were cured by adding a curing agent (0.1 wt%), and they were allowed to solidify for 2 h. The curing agent, NS625B (Nippon Steel), was used as a cross-linking agent and determined the amorphous MRE matrix properties. The curing pressure was consistently 12.963 kPa throughout the process, and there was no evidence of gravity segregation. The curing procedure for isotropic MRE was carried out in a closed cylindrical mold, with a 50 mm diameter and a 1 mm thickness of the circular sinking section inside the mold under the off-state condition. Based on previously published results [4,8,17,28], the application of a 0.1 wt% curing agent, the ready mixture of 30 wt% matrices, and 70 wt% CIPs were chosen for this study. Finally, a circular disc sample of the MRE was punched out using a hollow-hole punch tool from the original prepared MRE disc sheet to the required diameter of 20 mm and nominal thickness of 1 mm, and it had a 20% less surface-to-volume ratio than that in the previous related study [21].

### 2.2. Strain Localization and Morphological Observation

The elastic performance of the MRE under dynamic shear stress relaxation was investigated using an oscillation parallel plate rheometer operating in the oscillatory shear mode (Physica MCR 302, Anton Paar Company, Graz, Austria). The temperature was set to 25 °C, which was maintained and controlled by the rheometer’s Viscotherm VT2, Anton Paar. The strain created by the oscillation of the rotating disc parallel plate was then set to a constant 0.01%, with a frequency of 1 Hz. The strain value was chosen within the elastic region known as the linear viscoelastic region (LVE). The strain level of 0.01% was determined in this study based on our initial rheological study of the sample and with the intention of understanding the structure at equilibrium, which is a good indication of how the material may respond; however, the material slightly shifts away from equilibrium at low deformation. Following durability testing, samples were prepared for microstructure analyses by cutting the sample (along the edges of the MRE) into segments with an approximate surface proportion of 1 mm × 10 mm. The surfaces of the cut samples were sputtered-coated before being examined with a field-emission scanning electron microscope (FESEM, JSM-7800F Prime, JOEL, Tokyo, Japan) set to 1 kV accelerating voltage. The morphological studies on the MRE sample were further studied utilizing a tapping-mode AFM analysis on a NanoWizard 3, NanoOptics AFM (JPK Instruments, Berlin, Germany) with a nanosensor tapping-mode monolithic-silicon AFM probe-type single-beam cantilever supplied by BudgetSensors, Sofia, Bulgaria.

## 3. Results and Discussion

### 3.1. Shear Band Analysis Using FESEM

The micrographs of the MRE sample morphology taken with an FESEM revealed a considerable variation associated with the distinctive CIP arrangement and distribution. There was no discernible defective surface characteristic identified in relation to the effect of the curing process. Figure 1 of the FESEM image depicts the typical surface condition of the isotropic MRE containing 70 wt% CIPs. The CIP–silicon matrix interface was fully incorporated and consistent throughout the sample. There were no obvious signs of excessive voids in the matrix regions. The typical micrograph at a higher magnification (×2000) in Figure 1b reveals that the matrix and particles firmly bonded, with no evidence of agglomeration or individual voids. These physical results demonstrate that choosing 70% CIPs as the optimal proportion for the MRE was consistent with previous research [29] on similar materials. This ratio was also used in most MRE studies, as detailed in the previous comprehensive review [5].

In the case illustrated in Figure 2a, shear stress, τ*,* can be written as follows:(1)τ=P/A
where the load, P, is applied transversely to the specimen per unit area, A. The strain, γ, associated with the shear stress, τ, is denoted as a deformation, δ, per unit length, *h,* as follows:(2)γ=δ/h

Figure 2b depicts the stresses, τ, and deformations, δ, caused by torsional shear in the MRE samples, which explains the stress analysis method used in this study. As a result, the geometry of the MRE-sample-specific deformation can be quantified. Consider an increase in the thickness, *dz*_2_ < *dz*_1_, of the sample, in which the top oscillates relative to the stationary bottom due to an increase in the angle, dθ. The deformation also known as the relative tangential displacement δ of the top of a vertical line drawn at r from the center is then
(3)δ=r dθ

Because the geometry of the deformation corresponds exactly to the earlier description of shear strain in Equation (2), the following can be written:(4)γ=δ/dz=r dθ/dz

In the stress relaxation durability test, the strain was fixed at 0.01%, the distance from the sample’s center was fixed at *r* = 10 mm, and the decrease in the thickness *dz*_2_ < *dz*_1_ of the sample resulted in a decrease in relative tangential displacement, *δ*. Therefore, this geometrical deformation consideration demonstrates the volumetric diminution in the affected area during the durability test in comparison to the thicker sample, limiting the stress distribution. Continuous shear stress applied rapidly at a fixed deformation causes structure degradation, such as molecular relaxation, cross-link disentanglement, and structural rearrangement [4]. Since stress relaxation happens at the molecular level and involves a change in the molecular chain arrangement, all these mechanisms contribute to the deterioration of the sample stress relaxation’s resistance property. Stress distribution with a low surface-to-volume ratio causes more permanent molecular slippage, and the cross-linkages of amorphous chains are broken due to uneasy reconfiguration within this confined and limited dimension.

Figure 3 shows the condition of the MRE sample after the durability evaluation, particularly after the 115,000-cycle test duration established previously in a related study [21]. Following the durability performance evaluation, a preliminary observation of the cut MRE micrography using FESEM, in general, revealed more intriguing morphology, reflecting the MRE’s narrower dimensions and less volumetrically suffered area than previously reported [21]. The cut area was chosen based on the torsional shear mathematical theory, which states that a sample’s shear center point is not precipitated by any torsional deformation (imaginary point); the torsional deformation was then compared to the shearing acting on the sample’s edge, which contributed to the greatest angular displacement. The shear band thickness or breadth was found to be larger, ranging from 1 µm to nearly 2 µm, whereas a previous study [4] found shear bands deformed with a width of less than 1 µm. Given the existing literature [6,30] on stress relaxation durability studies, this could be an improved and significant discovery. A previous discovery revealed similar phenomena in some parts of this current investigation. However, it is clear that the pattern of the phenomenon discovered in the current study is distinct. Stress relaxation phenomena in MRE have been theorized to occur via a variety of mechanisms [4], including microplasticity and the formation of shear bands by localized strain. Depending on the nature of the MRE matrix’s molecular structure, the shear band deformed in the sample exhibits a variety of patterns and sizes. The formation of shear bands due to strain localization becomes much more facile as the surface-to-volume ratio decreases. As a result, however, the shear band only deforms in the matrix with no sign of particle influence rather than simply orienting the shear band formation. Aside from this current study, shear band images on MRE have been scarcely published in the literature to the best of the authors’ knowledge. These findings lend support to the theory that shear bands are composed of permanently microplastically stretched amorphous molecular chain structures. To emphasize, this could occur even in the elastic region and worsen as the surface-to-volume ratio decreases, as schematically proposed in Figure 4.

### 3.2. Morphological Analysis Using AFM

The extensive studies on the use of AFM in investigating these stress relaxation and strain localization phenomena in MRE constitute the technical contribution of this work. This study’s findings will be useful for comprehending the microplasticity behavior in the formation of the shear band within a material’s elastic region. Figure 5 depicts the morphological features of inter-domains, as well as the microphase separation between soft and hard localized domain regions. This was accomplished using the AFM taping method. This method systematically transforms the phase offset of the harder domain to appear lighter in color, while the softer domain appears darker. The shear band, as shown in Figure 5b, is darker than the surrounding matrix, which was not subjected to microplasticity. This indicated that a localized area of the shear band softened due to microplasticity. At a line rate of 0.157 Hz, the scanned area images (35 × 40 µm) had a resolution of 512 × 512 pixels. Another possible explanation for this difference is the stress relaxation mechanism, which softened the molecular chain over time. However, as the durability duration continues to increase, the localized area of the shear band may grow in size. As indicated by the dashed ovals in Figure 6, the zone closest to the shear band underwent phase separation, resulting in microplasticity and an increase in the size of the shear band. A close examination of the shear band revealed the outer region of the elastic matrix domain. Splitting the softened chains and adhering to the harder domains of the elastic matrix resulted in the microphase separation of the elastic and microplastic deformations. The AFM cross-sectioned image-measuring features were used to determine the separated spacing. The deformed shear bands with thicknesses of less than 1 µm were measured, and the smaller range was found to be populated within the sample’s outer edge area. Additionally, the use of phase imaging AFM allowed us to distinguish and identify the hard and soft domains of the scanned area. By the end of the test, the observation of a thicker size shear band indicated that the strain was localized.

### 3.3. Microstructural Degradation and Stress Performance

The declining performance when stress was applied during the continuous shearing process revealed the effect of microstructural degradation. Understanding the stress pattern using evaluations enables the detection of strain localization onset behavior, which refers to the start or initiation of microstructural deterioration. Figure 7 depicts the shear stress performance at 7500 cycles, and it indicates that performance decreased as the number of cycles increased. The degradation was leisurely slow at the beginning of the 5000 cycles until it reached a further drop with a sudden descent of approximately 300% gradient difference. Under both conditions, the correlation in the linear regression plot shows a decreasing slope gradient. The degradation had only begun at this point, and the total stress reduction during this period was only about 1%. However, further investigation over a longer time period revealed more severe degradation, as shown in Figure 3 in the previous section.

The sample that reacted to minor changes in its original condition was studied at this early stage. Based on this finding, it can be concluded that a slow degradation process occurred in this material during the initiation of the resilience condition, and the degradation changed as the test duration increased. Specifically, the molecular chain structure of this material is capable of sustaining the elastic performance, and improved sustainability can be achieved through the response to magnetic stimulation, which effectively alters the elastic performance. MRE conceptualization consists of a molecular chain structure and the cross-linkages of cured MRE, as shown in Figure 8. The secondary interaction in this study was the restricted region between the amorphous matrix and the CIP crystalline molecular chain structure. The CIPs with rigid spheres in the crystalline molecular structure were randomly embedded in the matrix, resulting in them molecularly embracing a portion of the enclosed matrix, which is known as secondary interaction. MRE’s elasticity and deformation limits are determined by the cross-linkages formed during the curing process. Such cross-linkages are a critical structural factor, as they impart high elastic properties while also stiffening the MRE system to prevent a viscous dominant response [4]. As a result of the constant shear stress by the MRE, the stress was distributed over the molecular structure, disentangling the chains and causing molecular slippage. As more slippages and cross-link losses occur in the MRE, the molecular structure becomes insufficient to be reconfigured, resulting in permanent deformation, as illustrated in Figure 8.

As shown, prior to the durability test, the matrix’s amorphous molecular structures were un-stretched with the initial cross-linkage position (red color) generated after the curing process. The CIPs were embedded in the matrix amorphous molecular structure with rigid spheres in the crystalline molecular structure at random. During shear, the velocity gradient that developed across the sample was greater at the most outer section and decreased toward the center. The shear band development and its mechanism significantly impacted the shear profile generated across the sample in this study. The illustrated figure depicts the mechanism of shear deformation in the sample’s center following durability testing. The stretched region was just below the localized elastic limits among the entangled amorphous molecular chains during the initial stress relaxation process, and the cross-linkages were then reversibly aligned. At this point, no permanent deformation occurred during the stretching. As a result, no visible marks could be seen. However, the constant shear load repeatedly softened the chains, eventually pushing them over the localized elastic limit. As a result, the plastic flow occurred in the inner portion of the molecular chain, just beyond the localized limit. The fluctuating stress that developed in the amorphous molecular chain was then attributed to chain deformation and rupture, as well as cross-linking at the soft dominated domain. Shear band deformation (yellow line) corresponded to molecular chain disentanglement, cross-linkage breaking, and the sequence of molecular slip events that contributed to shear band homogeneity.

## 4. Conclusions

The purpose of this study was to determine the morphological effects of stress relaxation and the surface-to-volume ratio on MRE durability. According to the analysis results, the degradation pattern of the MRE samples varied toward stress relaxation within their elastic region and was influenced by the surface-to-volume ratio. This forensic investigation provided a critical morphological evaluation of the material quality and elastic performance. A complex and critical microstructural evaluation was established to determine the shear band constituted by strain localization during stress relaxation. The current study on the morphological effects of strain localization in the elastic region of MRE resulted in a more comprehensive understanding of the failure that occurs during stress relaxation durability performance. According to the current findings, the geometry effect (surface-to-volume ratio) of the sample plays a critical role in MRE elastically sustaining during stress relaxation. The morphological shreds of evidence established a correlation between the effect of the surface-to-volume ratio and stress relaxation durability performance. The use of AFM in the morphological analysis and the localized strain deformation zone contributes significantly to the current literature. The shear band was responsible for the MRE’s declining elastic performance, which could be a precursor to rupture. These findings could be used by design engineers and material scientists to select the appropriate material performance for various applications, thereby improving device capabilities and lifetimes. Material sustainability is critical to ensuring long-term durability and service life applications. This research could be used to optimize physical design parameters that are susceptible to durability performance under fixed small strain or deformation modes over long periods of time. In terms of industrial relevance, this research, for example, aids in understanding the effect of the geometrical design of the isolators and dampers used in adaptive vibration control in civil engineering. On a larger scale, estimating the life and degradation behavior of MRE products under all conditions is extremely useful, and the current study contributes to future industrial applications and technological advances in MRE durability evaluation. Even though the results achieved in this work are not immediately applicable to practical applications, they will be very useful in analyzing the failure or stress concentration of MRE applications, such as vibration isolators. In addition, to identify the effect of geometry (surface-to-volume ratio) on MRE durability, specifically under stress relation, an experimental test of MRE samples with different concentrations of curing agent needs to be explored.

## Figures and Tables

**Figure 1 materials-15-08565-f001:**
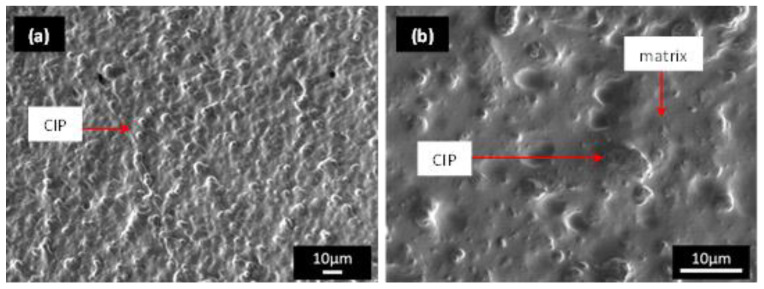
FESEM images of particle dispersion in an isotropic distribution of MRE at (**a**) low (×600) and (**b**) high (×2000) magnifications before the durability shear test.

**Figure 2 materials-15-08565-f002:**
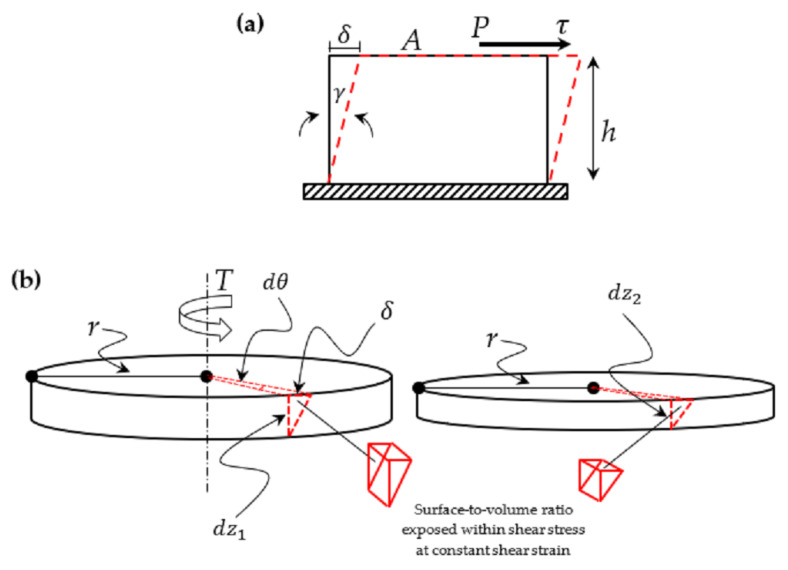
Illustration of shearing deformation by (**a**) the applied force and application area direction, and (**b**) geometrical deformation consideration by torsion and surface-to-volume comparison.

**Figure 3 materials-15-08565-f003:**
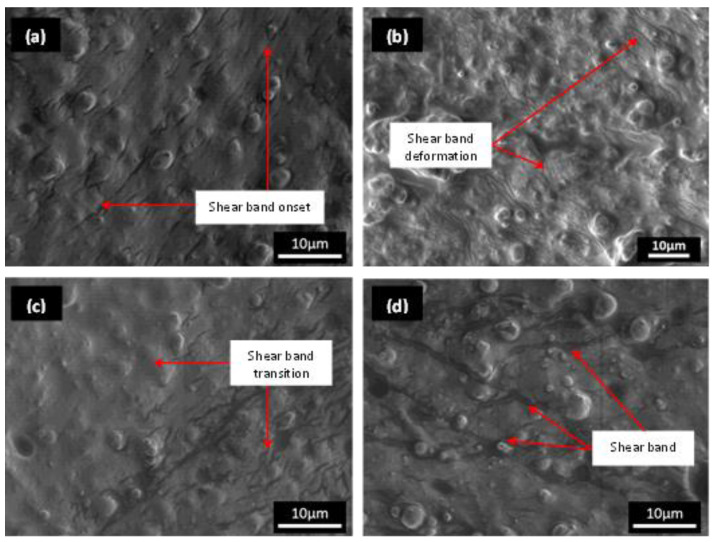
FESEM images of shear band deformation: (**a**) at the sample’s edge during the early test duration, (**b**) at the sample’s edge, (**c**) at the transition toward the center of the test sample, and (**d**) at the sample’s edge at the end of the test.

**Figure 4 materials-15-08565-f004:**
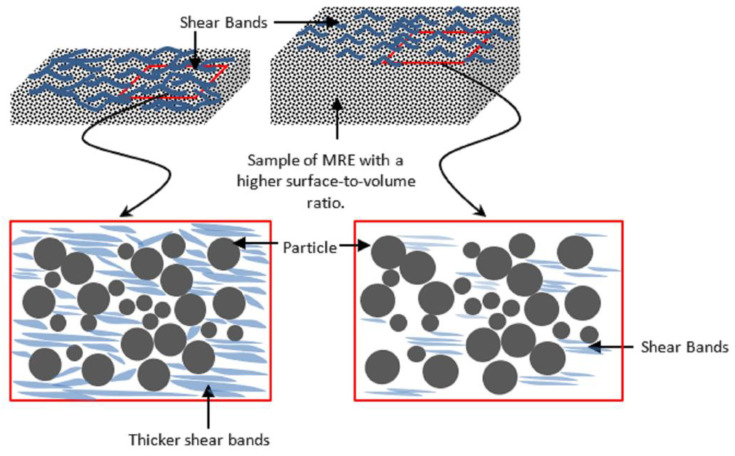
Schematic representation of the proposed mechanism for MRE under strain localization and shear band development.

**Figure 5 materials-15-08565-f005:**
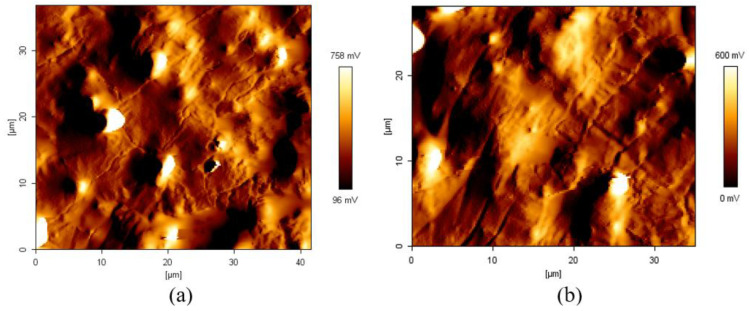
AFM images of the shear band deformation in the MRE matrix (**a**) during the early formation of the shear band and (**b**) after the durability evaluation, where shear band deformation with larger spacing was observed.

**Figure 6 materials-15-08565-f006:**
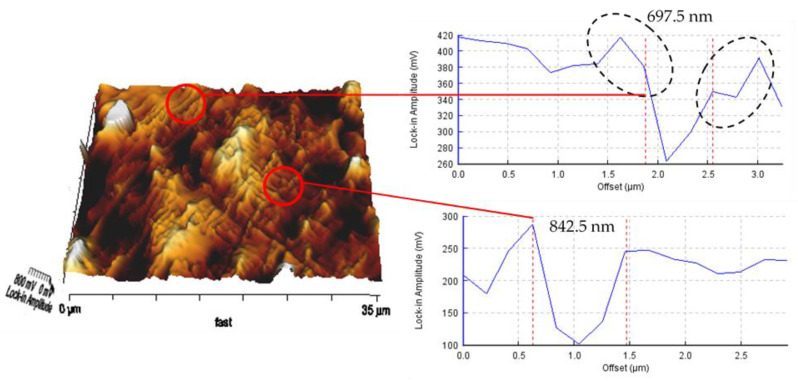
AFM three-dimensional amplitude micrograph of shear band deformation in the MRE matrix. The dashed ovals represent the shear band deformation zone.

**Figure 7 materials-15-08565-f007:**
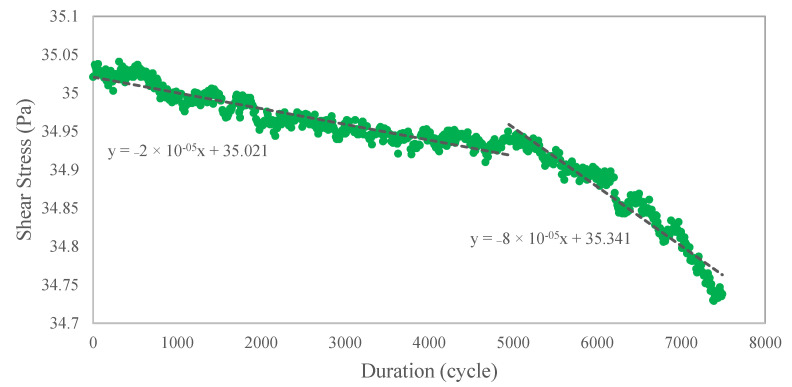
Shear stress performance over the test duration.

**Figure 8 materials-15-08565-f008:**
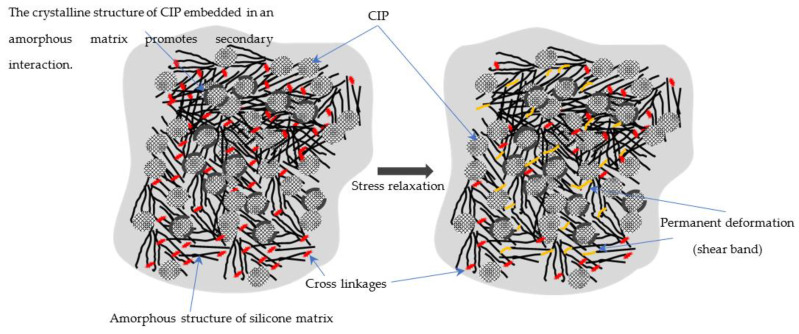
Schematic representation of CIPs, cross-linkages, and the matrix amorphous molecular chain undergoing stress relaxation resulting in permanent deformation.

## Data Availability

The raw/processed data required to reproduce these findings cannot be shared at this time, as the data also form part of an ongoing study. In the future, however, the raw data required to reproduce these findings will be available from the corresponding authors.

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
