# Peer review of "Morphological Effects of Strain Localization in the Elastic Region of Magnetorheological Elastomers"

_materials, 2022, doi:10.3390/ma15238565_

Round 1
Reviewer 1 Report (Previous Reviewer 2)
Although I see that the authors have incorporated an additional scheme of the performance of the CIP, it needs to describe better the actions that take place during the degradation process. Both figures in the scheme are similar, the difference is in the yellow "crosslinkings", but I continue to think that its needs a better explanation.
Other proposed request, like production of samples has not been answered, all manufacturing are based in some references, I think that in order to follow better the manuscript it need to expalin into samples production section.
I continue thinking that the manuscript is interesting, although I think it should go deeper into the magnetorheological behavior of these materials as they degrade due to stress relaxation. (i.e why is interesting for the industry, what is the role of the amount of CIP, the influence of curing agent, etc..)
I regret to say that from my point of view the manuscript has to be rejected.
Author Response
Please refer to the uploaded file.

Reviewer 2 Report (Previous Reviewer 1)
Most of the concerns were addressed. If accepted, please ensure all the text is in the same format/color.
I still have a minor concern about the SEM images. The authors mentioned: "It may appear similar, but it is not the same image. It has been captured in a similar area but represents a different image and location." I understand this statement. Figure 3b now seems similar to the inset in figure 1 (right-hand side) of the previous study. If it is a similar area, I interpret it as the same sample analyzed in the previous study.
The authors also addressed this in the paper: "Figure 3 shows the condition of the MRE after the durability evaluation, particularly 220 after the 115,000 cycles test duration established previously in a related study.[21]"
To avoid potential ethical concerns (the same sample analyzed but different studies and publications), I recommend moving figure 3 to the supplemental information. It is, however a suggestion.
Author Response
Please refer to the uploaded file.

Reviewer 3 Report (New Reviewer)
1. The ° should not be ^o (e.g. oC instead of °C)
2. 2.2. how was the linear viscoelastic limit determined? 0.01% seems extremely low. Are you sure it is not 0.01 (=1%)?
3. “The previous discovery has not revealed any similar phenomena. “ ??
4. The setup means that the strain is max. at the outer edges of the sample and 0 in the center. In this context, I do not quite understand what the authors wanted to say in the description of Fig. 3.
a. The shear bands I can see are way longer than 1-2 µm
b. What is meant by “whereas previous study [4] found shear bands deformed with less than 1 μm”? length of the shear band or deformation amplitude (which should be given in deformation not in length change)
c. The text (220-241) does not mention anything about the different sample locations tested. I would expect that the shear bands disappear in the center completely. If I would do this experiment myself, I would e.g. count the number or length of shear bands visible in one image at a magnification x and plot it vs. really experienced deformation (increasing from 0 in the center to the max. value at the outer edge.
5. What is the point about the discussion of amorphous regions (Fig. 4/5)? Silicones are all amorphous.
6. Fig. 3 – dark purple labels on black background are not well readable
7. Fig. 5 considering my comment 4, it should be clearly stated WHERE the sample that was characterized came from (e.g. from the outside of the sample). Furthermore, not only the the height profile but also mechanics should be shown locally. I expect shear bands to be significantly softer than the rest of the sample
8. Fig. 7 – 7500 cycles are shown but the experimental part talks about 115000 cycles. Please clarify.
9. Fig. 8: please show a DSC to prove that crystallinity ACTUALLY exists and how much fraction is there. As stated before, I have never heard of crystallizing silicones (at room temperature).
Overall, the topic is interesting, but the paper does not resemble a systematic study that can be understood without too much guesswork.
Round 2
Reviewer 1 Report (Previous Reviewer 2)
In my two previus reports I explain that my recomedation was REJECT, I follow in the same way and my recomendation in the 3 round and next rounds REJECT.
This manuscript is a resubmission of an earlier submission. The following is a list of the peer review reports and author responses from that submission.
Round 1
Reviewer 1 Report
Authors should include more detail about the curing agent used.
Section 3.3 needs to be enhanced. It is unclear how the authors applied magnetic stimulation in this study (methods). "pure sustainability was achieved through the response to magnetic stimulation, which effectively altered the elastic performance." Besides, there is only one data set on the shear stress vs. duration cycle. In this section and throughout the paper, the authors should clarify the wording cycles vs. seconds(abstract, results and discussion, and conclusion). The authors also correlate these results with figure 3, but it is unclear how these are compared, whether the test was performed under the same conditions, or at which time/cycle the imaging was taken.
An additional concern is that figure 3a (FESEM) seems to be a zoom-out of a previously published figure, found in https://doi.org/10.3390/ma14164384.
Moreover, the previous study has various similarities with the presented work, and the authors are encouraged to highlight and emphasize the new aspects of this work. SEM and AFM-phase imaging were also performed in the earlier study.
The conclusions section need to highlight the outcomes of the work. Authors should avoid conclusions such as "The findings also implied that the morphological characteristics that contributed to the shear band deformation observed in the system would be found in other viscoelastic material systems."
How can the authors conclude that these can be found in other viscoelastic materials when only a subset was run?
Reviewer 2 Report
The manuscript is interesting, although I think it should go deeper into the magnetorheological behavior of these materials as they degrade due to stress relaxation.
Other details that should also be highlighted; the aim is not clear and should be rewritten.
The Production of the sample section needs a better explanation (what curing agent is used, what other additives have been used, etc...).
The CIP carbonyl iron particle, some scheme would be necessary to visualize the kind of compound and the interaction that may exist between components
I regret to say that from my point of view the manuscript has to be rejected mainly due to its lack of depth.
Reviewer 3 Report
Dear Editor,
I have reviewed the article entitled Morphological Effects of Strain Localization in the Elastic Region of Magnetorheological Elastomers " by Mohd Aidy Faizal Johari, Saiful Amri Mazlan, Nur Azmah Nordin, Seung-Bok Choi, Siti Aishah Abdul Aziz, Shaari Daud, and Irfan Bahiuddin (Manuscript ID materials-1936290).
Here my comments:
In general, the article needs a major revision before resubmission: although the English is Ok, the manuscript is very technical, describing several phenomena, without providing sufficient support and explanations.
I would recommend an English revision since some phrases are incorrect.
The abstract does not show the research and manuscript highlights, only a brief description.
The introduction part is too short, without showing the relevancy of the research and explanation of the materials and methods selection.
The preparation method needs more details regarding the “traditional method” used (line 108). What was the curing conditions? Room temperature? Were the samples post-cured? Did the authors removed impurities using solvent rinsing? Etc.
It is better using freeze fracture technique for cutting of the samples, otherwise the cutting may affect the sample morphology.
The viscosity measurement comprises flow (continuous) and dynamic (oscillation) experiments, where viscosity, G’ and G” can be evaluated, respectively.
It is recommended adding different CIP concentration, as the rheology is comparative in this case. Also, did the authors used blank samples, of silicon rubber without CIP?
Did the authors tried different concentrations for the curing agent? The crosslinking degree have dramatic effect on the matrix mechanical and viscoelastic properties.
Figures 1 and 3 (FESEM images)- please add the image scale bar. Also, please add the instrument deformation direction (I assume it is perpendicular to the shear band).
I suggest using image analysis software for more accurate assessment of the shear band sizes and statistics. Do you have larger and clearer HRSEM images? Can you add images at different time intervals?
The AFM images can be improved. It can be compared to the images analysis suggested earlier.
There is still work to be done regarding the context. The manuscript is somewhat shallow. The discussion mainly describes the results and lacks explanations. It is recommended that the authors will review the text before re-submission.
Major revision is needed before resubmitting the manuscript.